# The Role of Neural Network for the Detection of Parkinson’s Disease: A Scoping Review

**DOI:** 10.3390/healthcare9060740

**Published:** 2021-06-16

**Authors:** Mahmood Saleh Alzubaidi, Uzair Shah, Haider Dhia Zubaydi, Khalid Dolaat, Alaa A. Abd-Alrazaq, Arfan Ahmed, Mowafa Househ

**Affiliations:** 1College of Science and Engineering, Hamad Bin Khalifa University, Doha 53, Qatar; uzsh31989@hbku.edu.qa (U.S.); khdo31645@hbku.edu.qa (K.D.); aabdalrazaq@hbku.edu.qa (A.A.A.-A.); arahmed@hbku.edu.qa (A.A.); 2National Advanced IPv6 Centre, Universiti Sains Malaysia, Gelugor 11800, Malaysia; haidardhia@yahoo.com

**Keywords:** Parkinson’s disease, neural network, deep learning, classification

## Abstract

*Background:* Parkinson’s Disease (PD) is a chronic neurodegenerative disorder that has been ranked second after Alzheimer’s disease worldwide. Early diagnosis of PD is crucial to combat against PD to allow patients to deal with it properly. However, there is no medical test(s) available to diagnose PD conclusively. Therefore, computer-aided diagnosis (CAD) systems offered a better solution to make the necessary data-driven decisions and assist the physician. Numerous studies were conducted to propose CAD to diagnose PD in the early stages. No comprehensive reviews have been conducted to summarize the role of AI tools to combat PD. *Ob**jective:* The study aimed to explore and summarize the applications of neural networks to diagnose PD. *Methods:* PRISMA Extension for Scoping Reviews (PRISMA-ScR) was followed to conduct this scoping review. To identify the relevant studies, both medical databases (e.g., PubMed) and technical databases (IEEE) were searched. Three reviewers carried out the study selection and extracted the data from the included studies independently. Then, the narrative approach was adopted to synthesis the extracted data. *Results:* Out of 1061 studies, 91 studies satisfied the eligibility criteria in this review. About half of the included studies have implemented artificial neural networks to diagnose PD. Numerous studies included focused on the freezing of gait (FoG). Biomedical voice and signal datasets were the most commonly used data types to develop and validate these models. However, MRI- and CT-scan images were also utilized in the included studies. *Conclusion*: Neural networks play an integral and substantial role in combating PD. Many possible applications of neural networks were identified in this review, however, most of them are limited up to research purposes.

## 1. Introduction

### 1.1. Background

The human brain is the primary controller part of the human body. Any minor damage to any of its parts will severely affect other organs—one of its adverse effects is Parkinson’s disease (PD) [1]. “PD is a chronic and progressive neurodegenerative disease” [2], and it occurs mainly in people over 50 years old [3]. Its symptoms start slowly and increase over time. PD symptoms are characterized such as motor and nonmotor [4]. Motor symptoms include movement disorders, shaking, walking issues [5], stiffness, and postural instability [6], while nonmotor symptoms including cognitive dysfunction, mood disorder [7], depression, and anxiety [8].

Parkinson’s is the second worse neurodegenerative disease worldwide after Alzheimer’s disease. In 2019, its incident rate ranged from 40.37 to 53.89 per 100,000 population per year in the US alone [9]. Diagnosis of PD in an early stage is an important issue to mitigate its complications. However, no medical test is available to diagnose it in the early stages conclusively. In a traditional clinical setup, the physician will ask the patient to perform some mental and physical tasks (e.g., moving and walking around) [10] or take the magnetic resonance imaging (MRI) and/or Positron emission tomography–computed tomography (PET/CT) scan of the brain. However, it is challenging to differentiate PD from other neurological disorders, and it depends on the radiologist’s experience to distinguish and identify it precisely. Therefore, a computer-aided diagnosis (CAD) system helps the radiologist interpret MRI scans. In 2003, the authors of [7] made a CAD system to monitor body acceleration to detect the freezing of gait in PD patients.

Several studies were conducted to implement machine learning approaches to detect PD and differentiate it from other common neurological diseases. Feature engineering is the difficult part of deploying such systems, and it is expensive to identify the relevant features in the data. When automatic feature extraction methods and techniques (CNN, RNN) were proposed, most researchers used deep learning and neural network to detect PD due to automatic feature extraction, learning more complex patterns, and high accuracy. Therefore, this scoping review aims to explore and summarize the applications of deep learning and neural network in PD diagnosis.

### 1.2. Research Problem and Objectives

The scope of this paper is limited to the detection of Parkinson’s disease (PD) in the early stage using neural networks. The patient dataset such as electronic health record (EHR) and medical image can be analyzed using neural network (NN) features; in particular, patient’s data can undergo many processes; analysis, segmentation, augmentation, scaling, normalization, sampling, aggregation, and sifting, in order to obtain accurate prediction that assists healthcare ecosystem and stakeholders in the healthcare domain. Many studies have been recently conducted to address and propose a solution to mitigate and prevent neurodegenerative disorders such as PD. However, most of these studies and research are dispersed. Therefore, summarizing NN technologies’ involvement in resolving challenges related to PD is needed; an appropriate summarization allows new researchers to understand the current role of neural networks against PD. It will open new opportunities for researchers to have the necessary base that allows them to build on instead of starting from ground zero.

Many studies have been carried out to cover AI techniques that have been used to mitigate and prevent PD [11,12,13,14]. These approaches are conducted in reviews or surveys that generally focus on artificial intelligence (AI) applications such as patient diagnosis, epidemiological monitoring, and drug and vaccine discovery [15]. Nevertheless, a massive number of research papers are constantly being published, which has overwhelmed electronic databases. Therefore, it is necessary to carry out an updated review that focuses on the uses of neural networks in PD prevention.

This review aims to identify and illustrate neural network technology’s role in detecting PD early, based on the following aspects: (1) identifying the role of neural networks in PD detection, (2) highlighting the recent algorithms applied on PD datasets, (3) observing dataset types, (4) categorizing the type of PD based on symptoms, (5) investigating the best results achieved by the research community, and (6) providing a recommendation for researchers and healthcare individuals. The outcome can be used in the healthcare sector as guidance for developers who consider neural network’s utilization to improve the public health capability as a response to PD.

## 2. Methodology

We carried out a scoping review to explore the evidence on neural network’s application in diagnosing Parkinson’s disease in a structured manner. In this section, we listed the details of the adopted methodology to conduct this review. For this purpose, PRISMA Extension for Scoping Reviews (PRISMA-ScR) [16] was used for this scoping review.

### 2.1. Search Strategy

#### 2.1.1. Search Sources

We selected five bibliographic databases (PubMed, IEEE, ACM, ScienceDirect, and Google Scholar) to retrieve the research studies relevant to the topic. We scanned only 100 articles from Google Scholar; these articles were chosen after scanning based on their relevance to fit this paper. The backward and forward reference checking lists were not performed due to the sufficient number of included studies. The search process was performed from 24 February to 1 March 2021.

#### 2.1.2. Search Terms

In the present review, we considered two different search terms based on population and intervention. Given the population of “Parkinson’s disease” and intervention of “deep learning”, the search strategy was conducted as follows: ((“Parkinson’s disease” OR “Parkinson*” OR “Parkinsonism” OR “paralysis agitans” OR “shaking palsy”) AND (“ artificial intelligence* “ OR “ machine learning” OR “ neural network*” OR “ deep learning” OR “natural language processing” OR “neural network*” OR “supervised learning” OR “unsupervised learning” OR “ensemble learning” OR “reinforcement learning”)) total retrieved studies in (Appendix A).

### 2.2. Study Eligibility Criteria

This study aims to summarize and review the application/use of deep learning, particularly in diagnosing Parkinson’s disease. Therefore, only the following studies were eligible to satisfy the below criteria: a deep learning approach or technique introduced or developed that primarily focused on diagnosing Parkinson’s disease. Further, some constraints on the types of publication and the language of the studies were made. Only studies published in English between 2018 and 2021 are selected, and only peer-reviewed articles, conference proceedings, reports, theses, dissertations were admitted. Reviews, conference abstracts, commentaries, proposals, editorials were excluded. The details of exclusion and inclusion for study selection are listed in Table 1.

### 2.3. Study Selection

The study selection process was conducted in two stages (screening title and abstracts of retrieved studies and screening full text of the studies selected in the first stage). In the first stage, the first reviewer, MA, independently screened all the retrieved studies’ titles and abstracts; due to time constraints, the second reviewer, US, and the third reviewer, KD, reviewed the first half and second half of the complete set of articles, respectively. The Rayyan software, a web-based systematic review tool, was employed for screening title and abstract [17]. In the second stage, the first reviewer, MA, performed the first stage’s full-text screening of the identified studies. Any disagreement between reviewers was resolved through consensus and discussion.

### 2.4. Data Extraction and Data Synthesis

To extract the study-specific information and data, an extraction form was created and tested by eight included studies (Appendix B). MA and US undertook the data extraction, and the data were extracted to the excel sheet to summarize the following: general characteristic of included studies (e.g., country, types, and year of publication), aim/purpose of the study, type of Parkinson’s disease, branch/type neural network, type of validation, performance metrics, the dataset used to train and test the model, number of Parkinson’s and healthy samples, type of dataset, size of the dataset, data collection device or sensor, and dataset source. We used the narrative approach to synthesis the extracted data.

## 3. Results

### 3.1. Search Results

In total, 1061 studies were retrieved by searching through 5 recognized E-Databases. Then, 190 (17.90%) were removed due to duplication, while 871 (82.09%) went through title and abstract screening; in this screening, we excluded 598 (56.36%) studies due to various reasons, as shown in Figure 1. The remaining 273 (25.73%) studies went through the full-text screening, and 181 (17.05%) studies were excluded, as detailed in Figure 1. In total, 91(8.67%) studies were included in this review.

### 3.2. General Description of the Included Studies

As shown in Table 2, the included citations were published in more than 30 different countries, as shown in Figure 2, about 13 studies from the US (14.13%), followed by 9 studies from China and India (9.78%) (Figure 3). This shows that numerous papers were published in the last 3 years; for instance, 30 papers (32.60%) were published in 2019 and 2020. More than half (56.2%) of the included studies were conference papers. However, most conference papers (*n* = 18) were published in 2018, and 2020, respectively, and only (*n* = 16) conferences article were reported in 2019. In addition, (*n* = 39) journal articles were published in last few years: (*n* = 10) in 2018; (*n* = 14) in 2019; (*n* = 12) in 2020; and (*n* = 3) in 2021.

### 3.3. Description of Detection Techniques

The study’s primary aim is to investigate the role of neural networks in the diagnosis of PD. We classified neural networks into five well-known algorithms used in the included studies: CNNs, RNNs, FNNs, ANNs, and other NNs. Around half of the included studies used convolution neural networks (*n* = 37); afterward, other neural networks (*n* = 31) were implemented in the included studies, followed by artificial neural networks (ANNs) (*n* = 10), recurrent neural networks (RNNs) (*n* = 9), and fuzzy neural networks (FNNs), as shown in Table 3. In the end, the most imitated neural network architecture in the included studies was LSTM (*n* = 11) [6,34,36,38,40,65,70,74,77,80,83], VGG (*n* = 3) [18,27,58], and DNN (*n* = 6) [34,35,60,91,92,103]. Recently, with the developments of new techniques such as convolutional neural network [101] and transfer learning [63], deep learning gained significant advances in the computer vision tasks, e.g., ImageNet [77]. Therefore, most of the studies used different imaging data to diagnose PD, such as MRI (*n* = 12) [41,47,54,56,58,66,72,78,82,86,90,95] and handwritten images (*n* = 9) [3,19,25,30,69,75,101,102], as well as PET and CT imaging (*n* = 6) [28,59,67,71,88,90] and DaTscan imaging (*n* = 4) [54,76,99,103]. However, CNN and transfer learning techniques were not limited to imaging data; they also learn complex features from voices and signal data [29]. Numerous studies used the biomedical voice (*n* = 21) [4,6,22,23,29,33,44,48,50,52,53,55,60,61,73,74,84,93,100,104,105] and biometric signal (*n* = 14) [26,31,34,36,45,46,57,62,64,65,68,89,96,98]; a few of the included studies used EEG and EMG signals (*n* = 5) [32,39,51,83,85].

As shown in Figure 4, some studies target specific symptoms of PD, such as freezing of gait, vocal impairment, and tremor disorder. A more limited number of included studies proposed a deep learning approach to detect tremor disorder (*n* = 5) and vocal impairment (*n* = 13). However, various studies used the deep learning technique to diagnosis PD (*n* = 50), in general, and freezing of gait (FoG) (*n* = 23), in particular.

As reported in Table 3, the neural network is divided into five main branches (CNN, RNN, ANN, FNN, NN); all types of subclassification techniques are listed as backbone model; moreover, we noticed that LSTM was heavily used in a different study (*n* = 11), followed by none deep learning classifier SVM (*n* = 8); however, we have reported SVM in this review because many studies used neural networks to perform data extraction, but the classification was handled by the machine learning classifier such as SVM; hence, DNN was used and reported in (*n* = 6), and a predefined model such as VGG was used in (*n* = 3); other types of algorithms that were used rarely depended on each of the studies’ design or achieved a remarkable result.

In most of the studies, the dataset was divided into three parts training, testing, and validation due to the limited number of studies that divided the datasets only into the training set and validation set, as presented in Table 3. We reported only the training and testing datasets. Furthermore, most of the experiments (*n* = 21) used ≥80%) volume of the training dataset, and (*n* = 9) used (≥70%). However, only few experiments provided less volume of the training dataset, as seen in (*n* = 5) used (≥60%) and (*n* = 3), (*n* = 1) used (≥50%), (≥40%), respectively. However, (*n* = 43) of the studies did not mention the volume of the training dataset. In addition, the volume of the testing dataset is not clarified in most of the studies; we noticed that (*n* = 53) did not specify the volume of the testing dataset that was used during the experiment; however, the volume of (≥20%) was mostly used in (*n* = 18), followed by (≥10%) that were mentioned in (*n* = 9), and the volume of (≥30%) was observed in (*n* = 6). The testing dataset is usually used in low volume, compared to the training dataset; however, we noticed that half of the dataset (≥50%) was used only in (*n* = 3). In addition, low volumes of testing dataset, i.e., (≥5%) and (≥40%), are reported in (*n* = 2) and (*n* = 1), respectively.

The validation method is highly considered in this review; we have reported all the studies’ validation mechanisms. The most common K-fold cross validation (K-FCV) methods used are the tenfold cross-validation, which was used in (*n* = 30), followed by fivefold cross-validation in (*n* = 12), whereas fewer K-FCV methods were reported as threefold cross-validation, fourfold cross-validation, sixfold cross-validation, sevenfold cross-validation, and eightfold cross-validation in (*n* = 2), (*n* = 1), (*n* = 1), (*n* = 1), and (*n* = 1), respectively. Furthermore, other validation methods such as LOSO, LOPO, LOOCV, and holdout were rarely used, and are reported in (*n* = 3), (*n* = 2), (*n* = 2), and (*n* = 1), respectively. However, (*n* = 36) did not mention any type of validation method within their experiments.

Various evaluation metrics used to check each model’s performance and accuracy are the most commonly used metrics to calculate the model’s efficiency in predicting the result based on the testing dataset. In (*n* = 57), the accuracy of the models was reported. On the other hand, along with the accuracy, other evaluation methods were used, such as recall/sensitivity that was reported in (*n* = 36), followed by specificity in (*n* = 24) and precision (*n* = 17); however, few studies (*n* = 8) used area under the curve (AUC) as an evaluation metric.

During summarization of all (*n* = 91) results, unfortunately, we did not come across any empirical validation/real-life implementation in any hospital. Moreover, from the (*n* = 91) studies, we only found one study that developed diagnosis software that identified any neurological disorders such as PD and that can be employed in the medical center [51].

### 3.4. Dataset Description

#### 3.4.1. Public Dataset

As discussed, an earlier total number of the public dataset (*n* = 57), Table 4, summarized the most used (*n* = 36) public available dataset sources and repositories (*n* = 36), e.g., Parkinson Progression Markers Initiative database (PPMI), UCI database repo, and PhysioNet; these were the most used datasets to develop and validate the AI models. Other public dataset sources used by the included studies were as follows: Kaggle, HandPD, DaphNet, the NTUA Parkinson Dataset, Neurovoz corpus, PC-GITA database, etc.

Table 4 only provides a sample of the public datasets used within the included studies. As seen, the number of males in the PD sample is higher than the number of females, and the number of males in healthy control is higher than the number of females in most cases. Furthermore, different types of hardware devices were used to collect the dataset; we have noticed that most of the data are in the form of images collected with different devices, starting from hospital imaging device including MRI, CT, DaTscan and ending with smartphone images that were used to capture handwriting or drawing of the PD samples (*n* = 28) and (*n* = 4) for recording video.

Biometric signal and time-sensor-based dataset were collected using the digital keyboard or sensor/accelerometer (*n* = 16) attached to the PD and healthy control sample or placed at a different angle to measure the severity of the freezing gait or the tremor. Moreover, devices such as a high-quality standalone microphone or smartphone were used to collect the biomedical voice dataset, and (*n* = 15) reported a public vocal dataset. Moreover, in the public dataset, only (*n* = 11) reported the gender of PD and healthy control sample, and only (*n* = 5) studies identified each sample’s mean age.

#### 3.4.2. Private Dataset

As mentioned, the earlier total number of private datasets (*n* = 31) is shown in Table 5. We summarized the dataset that was clearly explained within studies (*n* = 5). This dataset was collected and labeled in different entities such as hospitals, universities, and research centers. The number of PD and healthy control samples are reported, including gender. Table 5 only provides a sample of the private datasets used within the included studies. The number of males in the PD sample is higher than the number of females, whereas the number of females in health control is higher than the number of males. Furthermore, different types of hardware devices were used to collect the dataset; we have noticed that most of the data were in the form of images collected with different devices, starting from hospital imaging device including MRI, CT, DaTscan and ending with smartphone images that were used to capture handwriting or drawing of the PD samples (*n* = 11).

Biometric signal and time-serious-based dataset were collected using the digital keyboard or sensor/accelerometer (*n* = 14) attached to the PD and healthy control sample or placed at a different angle to measure the severity of the freezing gait or the tremor. Moreover, devices such as a high-quality standalone microphone or smartphone were used to collect the biomedical voice dataset, and (*n* = 6) reported a private vocal dataset. Moreover, in the private dataset, only (*n* = 4) reported the gender of PD and healthy control sample, and only (*n* = 4) studies identified each sample’s mean age.

## 4. Discussion

### 4.1. Principal Findings

Although this study focuses on identifying and addressing deep learning and neural network application to detect Parkinson’s disease in the early stage, we found some proposed models show promising results and can be employed in hospitals. This review provides recommendations for professional healthcare and researchers based on the included studies’ outcomes. Moreover, we noticed that five studies [21,37,49,55,81] used the Vertical Ground Reaction (VGRF) dataset, which was obtained from PhysioNet hub to train the classification models including fuzzy neural networks (FNNs), stacked 2D CNNs, deep neural networks (DNNs), artificial neural networks (ANNs), and neighborhood representation local binary pattern (NR-LBP). However, DNN in [49] surprisingly achieved outstanding results for early detection of PD using the VGRF dataset, compared to the other studies.

Furthermore, for imaging dataset including MRI, PET CT, and DaTSCAN were mainly obtained from Parkinson Progression Markers Initiative (PPMI) to train classifier, as seen in [20,28,41,47,59,66,67,76,82,86,88,90,94,95]; hence, among all studies, CNN in [20] and FNN in [28] achieved an outstanding result for image classification.

We found that most of the biomedical voice measurements dataset was obtained from the University of California (UCI) Irvine Machine Learning repository; in [53,84] and [23], the same dataset is used; however, 19 achieved outstanding result using the sequential model in a deep neural network for detection PD based on voice measurement. In [33,44], and [4], the same voice measurement datasets with 756 instances and 754 attributes were used to identify PD, and the autoencoder neural network in [33] achieved better results than other studies.

Electroencephalograph (EEG) dataset was obtained from a different source and used in five studies [32,38,51,83,85]. In [38,83], we found that long short-term memory (LSTM) achieved outstanding results, indicating the best option to deal with EEG data. On the other hand, seven studies [3,19,25,27,40,69,101,102] focused on the classification of handwriting image to identify PD in the early stage, and we found that outstanding results were achieved in ANN + SVM in [3], dual-path RNN (DPRNN) in [40], and CNN + Optimum-Path Forest (OPF) in [102], respectively.

As mentioned earlier, the detection of PD using a neural network is not an easier task than other types of diseases because PD symptoms (vocal disorder, tremor disorder, freezing gait disorder) are inconsistent, and it is difficult to collect data concerning the type of the device. Therefore, many public repositories mainly focus on collecting and process certain types of datasets. Moreover, based on our findings, we can conclude that the sequential model in DNN and autoencoder neural network proved to be suitable models for PD detection from speech. Moreover, DNN is recommended to identify PD from VGFR data. Additionally, CNN is still on top for medical image classification such as MRI, PET/CT, and DaTSCAN. Moreover, the FNN shows significant results in classifying a medical image. On the other hand, in regard to images of handwritings, we found that ANN with machine learning classifier SVM had a remarkable result for the identification of PD from handwriting.

Based on the findings of this review, we can highlight the most used repositories that contain PD public datasets for the research community as follows: (1) UCI Repository of Machine Learning Database, University of California; (2) PhysioNet Laboratory for Computational Physiology, Massachusetts Institute of Technology; (3) Parkinson’s Progression Markers Initiative (PPMI); (4) Pacific Parkinson’s Research Institute; and (5) Botucatu Medical School, São Paulo State University, Brazil.

### 4.2. Strengths and Limitations

#### 4.2.1. Strengths

This review covered deep learning neural network techniques used for PD detection regardless of the characteristics, country, and study design. We claim that this review is a comprehensive study of neural network approaches used for PD detection. It will help researchers to understand how neural network is used efficiently for detecting PD in early stages. Compared with other reviews [106,107,108] that do not focus on PD disease, this review is unique in its field because it describes and summarizes features of the identified neural network models, datasets, available repository, type of PD evaluation, validation, and research implication. Moreover, this review is different from the previously mentioned reviews by following the latest version of PRISMA-ScR [16]. Unlike other reviews, we retrieved the studies from the most popular computer science and healthcare database to determine the most relevant studies possible.

#### 4.2.2. Limitations

In the beginning, we carried out a primary search from 2015 to 2021 through the five selected databases, and we retrieved a massive number of studies. Therefore, we limited our search to the period between 2018 to 2021. Due to that, we may have missed some significant studies. Due to many studies that we included (*n* = 92), backward and forward reference checking was not performed in this review. PD is an extensive topic and divided into many types of diseases, including various symptoms. Therefore, we may have missed categorizing some diseases from a clinical perspective.

### 4.3. Practical and Research Implications

Although this review investigates the neural networks used to detect Parkinson’s disease (PD), some applications could significantly mitigate this neurodegenerative disorder. Nowadays, computer-aided diagnosis systems are essential because they are less time consuming and more user friendly. For example, the authors of [51] designed a GUI system that physicians may use for fast diagnosis of Parkinson’s disease in its early stages. Researchers can also use the system to continue their future research on disease diagnosis, especially neurodegenerative disorders. The system will show the patient’s disease progression and help clinicians monitor the disease in its early stages.

Furthermore, the system can differentiate between PD patients and healthy subjects and compare various parameters (EEG, EMG, MRI/PET scan). In both PD and control subjects, the model can detect the region of dopamine output in the substantia nigra. As a result, the proposed model would be a novel solution containing all of the PD detection parameters in a single window, which would be extremely useful for disease monitoring.

In the included studies [6,18,19,30,61,75,87,91,92,96,98,101], clinicians could obtain PD Patient data in telemonitoring using devices such as tablets and smartphones. It is a promising solution because they can increase monitoring frequency without putting a strain on professional resources during the COVID-19 pandemic. However, the cost of training and testing the detection algorithm on a smartphone was too high; thus, the results were measured on a remote server and then transferred to the computer.

Clinical studies can refer to a video recorded for the patient while performing physical activities such as a PD bed test. As mentioned, in [18,43,70,87], a neural network was able to identify the symptoms of PD through a video sample of the patient. In the future, the clinical studies may analyze any video recorded in the hospital for other patients, for example, during therapy sessions, and predict if this patient is suspected of having PD in the future.

## 5. Conclusions

This scoping review summarized studies by investigating the use of neural networks, specifically deep learning algorithms, for early diagnosis of PD based on various data collected from different public and private sources (91 studies), including medical image, biomedical voice, and sensor signal, for both PD and healthy control samples. Included studies were categorized into different groups based on the neural network model, type of PD symptoms, and type of dataset. Additionally, the most used dataset and best performance model were highlighted based on the detection of particular symptoms of PD in this review. All technical experiment methods were reported, including submodel, dataset volume, training, testing, evaluation metrics, and validation type. We indicated any real-time implementation used in each hospital or university setting, and based on this review, we recommended particular suggestions for healthcare professionals. Future work could be a meta-analysis to examine each study and provide a comprehensive comparison between them in terms of quality.

## Figures and Tables

**Figure 1 healthcare-09-00740-f001:**
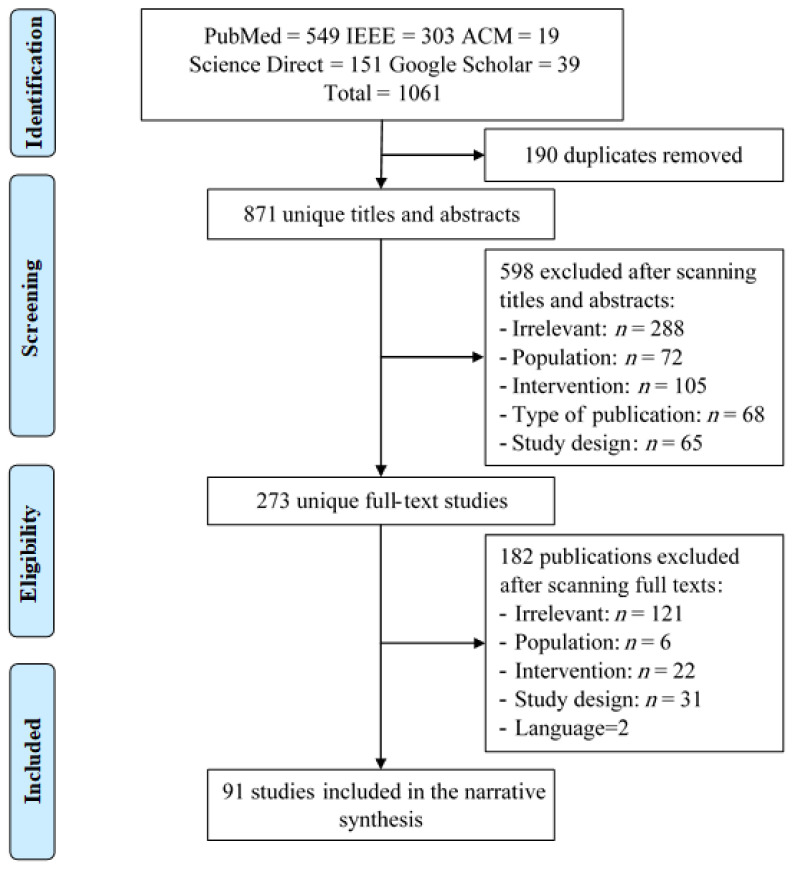
PRISMA chart.

**Figure 2 healthcare-09-00740-f002:**
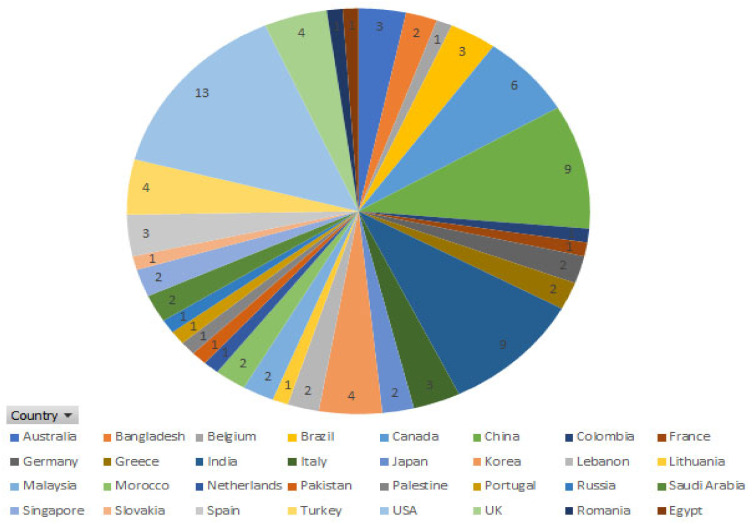
Number of publications for each country.

**Figure 3 healthcare-09-00740-f003:**
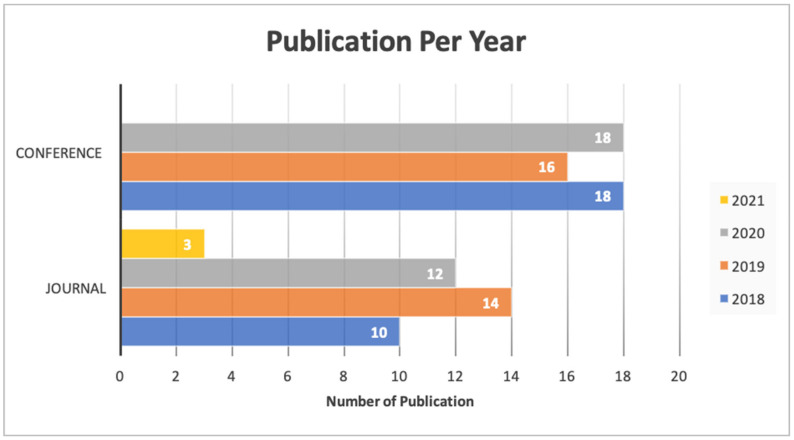
Type of publication and year.

**Figure 4 healthcare-09-00740-f004:**
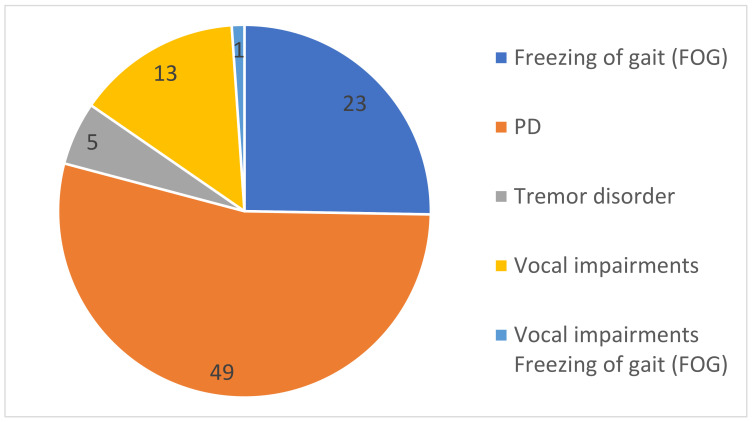
Different symptoms of Parkinson’s disease in the included studies.

**Table 1 healthcare-09-00740-t001:** Inclusion and exclusion criteria.

Criteria	Specified Criteria
**Inclusion**	Studies that aim to diagnose Parkinson’s using deep learning technique or approach Studies that published from 2018 onwards Empirical studies only Only written in English
**Exclusion**	Abstract Review including an overview, scoping review, etc. Non-English studies Non-peer-reviewed articles

**Table 2 healthcare-09-00740-t002:** General characteristics of the included studies (*n* = 91).

Characteristics	Studies, *n* (%)	Ref.
Year of publication	2021: 4 (4.34)	[6,18,19,20]
	2020: 30 (32.60)	[21,22,23,24,25,26,27,28,29,30,31,32,33,34,35,36,37,38,39,40,41,42,43,44,45,46,47,48,49,50,51]
	2019: 30 (32.60)	[4,52,53,54,55,56,57,58,59,60,61,62,63,64,65,66,67,68,69,70,71,72,73,74,75,76,77,78,79,80,81,82,83,84,85]
	2018: 28 (30.43)	[3,86,87,88,89,90,91,92,93,94,95,96,97,98,99,100,101,102,103,104,105]
Country	US: 13 (14.13)	[18,27,29,57,61,65,74,82,87,88,92,95,104]
	China: 9 (9.78)	[33,40,52,53,66,67,85,89,90]
	India: 9 (9.78)	[3,31,37,50,51,55,60,63,105]
	Canada: 6 (6.52)	[35,38,45,46,83,93]
	UK: 4 (5.43)	[48,58,62,103]
	Korea: 4 (4.34)	[30,41,56,98]
	Turkey: 4 (4.34)	[4,36,77,101]
	Brazil: 3 (3.26)	[75,97,102]
	Australia: 3 (3.26)	[20,42,100]
	Italy: 3 (3.26)	[21,49,96]
	Spain: 3 (3.26)	[76,91,94]
	Greece: 2(2.17)	[54,99]
	Bangladesh: 2 (2.17)	[44,59]
	Japan: 2 (2.17)	[6,72]
	Lebanon: 2 (2.17)	[68,69]
	Malaysia: 2 (2.17)	[39,84]
	Germany: 2 (2.17)	[71,79]
	Morocco: 2 (2.17)	[23,25]
	Saudi Arabia: 2 (2.17)	[28,80]
	Singapore: 2 (2.17)	[32,81]
	Belgium: 1 (1.08)	[43]
	Colombia: 1 (1.08)	[70]
	France: 1 (1.08)	[47]
	Lithuania: 1 (1.08)	[22]
	Netherlands: 1 (1.08)	[78]
	Pakistan: 1 (1.08)	[86]
	Palestine: 1 (1.08)	[73]
	Portugal: 1 (1.08)	[64]
	Russia: 1(1.08)	[26]
	Slovakia: 1 (1.08)	[19]
	Romania: 1 (1.08)	[24]
	Egypt: 1 (1.08)	[34]
Type of publication	Conference: 52 (56.52)	[3,18,21,22,23,24,25,26,27,29,30,31,36,44,45,46,47,48,49,50,51,53,54,55,56,57,58,59,60,61,63,64,65,66,68,69,70,71,74,77,79,81,82,83,84,85,86,87,88,89,92,95,96,99,100,101,102,103,104,105]
	Journal article: 39(42.39)	[4,6,19,20,28,32,33,34,35,37,38,39,40,41,42,43,52,62,67,72,73,75,76,78,80,90,91,93,94,97,98]

**Table 3 healthcare-09-00740-t003:** Description of PD detection techniques (*n* = 91).

Characteristics	Studies, *n*								
**Type of PD** **symptoms**	**PD: 49**		**FoG: 23**		Vocal impairments:13		Tremor disorder: 5		Vocal impairments and FoG:1
**Dataset Source**	**Public:57**				Private:31		NA: 3		
**Type of Dataset**	MRI: 12 DaTscan: 4 PET&CT images:6 Handwriting Images: 9		Biomedical Voice:21		Biometric signal: 14		EEG and EMG: 5		VGRF time series: 4 Video: 4
**Neural Network**	CNN: 37 RNN:9 ANN: 10 FNN:4 Other NN: 31								
**Model Backbone**	LSTM: 11 SVM: 8	DNN: 4 VGG: 4	Autoencoders (AE): 2 DCNN: 2 MLPs: 2	Inception v3: 1 AlexNet: = 1 RestNet: 1 U-Net:1	WGAN: 1 ASE: 1 SSAE: 1 LSVRC: 1 DNMLDM: 1 DPRNN: 1	LRNN: 1 MTL: 1 GCN: 1 GS-RNN: 1 NR-LBP: 1 TCN: 1	OPF: 1 FRP: 1 FCNN: 1 EFMMOneR: 1	Encoder-Decoder DBN: 1 MOGA: 1 BiLSTM: 1	SSM-PCA: 1 SNN: 1
**Training dataset Volume**	≥80%: 20		≥70%: 19		≥60%: 5	≥50%: 3	≥40%: 1		NA: 43
**Testing dataset Volume**	≥50%: 3		≥40%: 1		≥30%: 6	≥20%: 18	≥10%:8	≥5%: 2	NA: 53
**Validation Method**	10-FCV: 29		5-FCV: 12		LOSO: 3	LOPO: 2 LOOCV: 2 3-FCV: 2	4-FCV: 1 6-FCV: 1 7-FCV: 1 8-FCV: 1	Holdout: 1	NA: 36
**Evaluation Metrics**	Accuracy: 56		Recall/Sensitivity: 35		Specificity: 24		Precision: 16	F1-Score: 7	AUC: 8
**Developed software**	Diagnosis dashboard: 1								

**Table 4 healthcare-09-00740-t004:** Public dataset descriptions.

Dataset	Source/Host	Used Device/Sensor	Number of PD Patient		Number of Healthy Control		Ref.
			Male	Female	Male	Female	
Public	PhysioNet (*n* = 4)	16 sensors s under each foot 8 per foot	59	34	40	32	[21,49,55,81]
	The University of California, Irvine Machine Learning repository UCI (*n* = 10)	NA	84	40	23	41	[3,4,23,33,44,53,55,60,84,105]
	Neurovoz corpus	NA	32	20	27	29	[74]
	PPMI (Parkinson Progression Markers Initiative) database (*n* = 14)	MRI Machine	129		57		[20,28,41,47,59,66,67,76,82,86,88,90,94,95]
	The NTUA Parkinson Dataset (*n* = 1)	DaTscan and MRI Machine	55		23		[99]
	PC-GITA database (*n* = 1)	Professional audio card	25	25	25	25	[50]
	Department of Neurology in Cerrahpasa Faculty of Medicine, Istanbul University (*n* = 1)	Wacom Cintiq 12WX graphics tablet	57		15		[101]
	HandPD dataset Botucatu Medical School, São Paulo State University (*n* = 2)	Smartphone Camera	59	15	6	12	[19,102]
	Daphnet Dataset University of California, Irvine Machine Learning repository (*n* = 2)	sensor was attached to a belt and above the ankle and above the knee	7	3	NA	NA	[62,65]
	Parkinsons drawing spirals and waves Kaggle	Tablet for capture the drawing	27		28		[30,101]

**Table 5 healthcare-09-00740-t005:** Private dataset descriptions.

Dataset	Source/Host	Used Device/Sensor	Number of PD Patient		Number of Healthy Control		Ref.
			Male	Female	Male	Female	
Private	Wearable Bio mechatronics Laboratory at Western University	wearable assistive devices for suppressing tremor	13		NA	NA	[46]
	Pacific Parkinsons Research Centre (PPRC)”	wearable headset with 27 electrodes to capture the EEG signals.	10	10	11	9	[83]
	Hospital at Sun Yat-sen University	64-electrode Geodesic Sensor Net (Electrical Geodesics Inc.)	25	15	18	12	[85]
	RMIT University, Melbourne, Australia	Apple iPhone 6S plus^®^	41		40		[100]
	*n*/A	Digital software keyboard	18		15		[79]

## Data Availability

Not applicable.

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
