# Peer review of "The Role of Neural Network for the Detection of Parkinson’s Disease: A Scoping Review"

_healthcare, 2021, doi:10.3390/healthcare9060740_

Round 1
Reviewer 1 Report
The authors should discuss more intensively that neural networks are often a supporting technology to enable other methods (such as natural language processing). The presentation of results is not very transparent and not focused.
A discussion of diagnostic guidelines is missing.
The selection of literature portals is insufficient because many are missing, such as Wiley, SpringerLink, Taylor & Francis, Emerald, SAGE, Web of Science).
The underlying idea is good, but perhaps an extension to "artificial intelligence" would be more helpful.
Author Response
The authors should discuss more intensively that neural networks are often a supporting technology to enable other methods (such as natural language processing). The presentation of results is not very transparent and not focused.
For scoping review, we did mention briefly about neural network but following PRISMA standard we are not required to go in details. Based on our findings highlighted all technology that have been used but we did not come across natural language processing that used separately
A discussion of diagnostic guidelines is missing.
This paper many pages and 8000 words we just mention diagnostic guidelines as traditional method to diagnose Parkinson's Disease
The selection of literature portals is insufficient because many are missing, such as Wiley, SpringerLink, Taylor & Francis, Emerald, SAGE, Web of Science).
We have scanned most common database and we obtained 91 studies as final result, this number is sufficient to conduct this review, in future we may conduct other scoping review by scanning the mentioned database.
Reviewer 2 Report
In this review article, the authors aim to explore and summarize the applications of neural networks to diagnose Parkinson's disease.
PRISMA Extension for scoping review (PRISMA-ScR) was followed as a recommendation to conduct this scoping review article.
Neural networks are believed to play an integral role in combating Parkinson's disease.
The paper is well organized and, a clear objective has set. The authors have included the necessary background for the readers in the article.
The review of the state-of-the-art is sufficient and up-to-date.
Moreover, the authors organized the article according to the proper structure:1) Introduction 2) Materials and Methods 3) Results 4) Conclusions.
It is structured according to the following axes:
1) Identify Neural Network role for detection of Parkinson's disease 2) Highlight the recent algorithm applied on PD dataset: 3) Observe the type of dataset that used 4) Categorize the type of PD based on symptoms 5) Investigate the best result that achieved by research community 6) Provide a recommendation for researcher and clinical.
This review article has covered deep learning neural network techniques used for the detection of Parkinson's disease, regardless of the characteristics, country, and study design.
Its scope is limited to Neural Network's use to detect Parkinson's disease in the early stage.
The main weaknesses and corrections needed in the text are the following:
1)The use of English is mediocre. There are several grammatical, syntactic, expressive and punctuation errors.
Writers need to redefine their writing style and adapt to a high-level scientific journal.
2)The Conclusions section should be more advanced and needs a better connection with the review results(numeric), especially as there is a potential for the issue discussed in this article.
3)I suggest the authors quote next to each reference the corresponding doi.
In addition, a good idea would be: In the text by clicking on the reference number to lead the reader directly to the corresponding reference.
[1] M. Alissa, “Parkinson ’ s Disease Diagnosis Using Deep Learning,” 2018.
Where has it been published?
4)I would like to emphasize that the authors have not used references from the MDPI journals.
There are journals like Applied Sciences, Brain Sciences, Diagnostics, International Journal of Environmental Research and Public Health, Medicina and Healthcare that have published articles in the subject area you are looking at with solid and credible approaches.
Author Response
1)The use of English is mediocre. There are several grammatical, syntactic, expressive and punctuation errors.
Writers need to redefine their writing style and adapt to a high-level scientific journal.
The paper has been properly edited to eliminate any errors. Further language editing will be done before the final submission.
2)The Conclusions section should be more advanced and needs a better connection with the review results(numeric), especially as there is a potential for the issue discussed in this article.
The conclusion section has been properly edited to create a better connection with the review results.
3)I suggest the authors quote next to each reference the corresponding DOI.
In addition, a good idea would be: In the text by clicking on the reference number to lead the reader directly to the corresponding reference.
The Corresponding DOI for each reference has been added.
We also followed the recommendation to make all reference numbers clickable, now the reader will be directed to the corresponding reference when clicking any reference number.
[1] M. Alissa, “Parkinson ’ s Disease Diagnosis Using Deep Learning,” 2018.
Where has it been published?
The reference has been edited in a proper manner.
4)I would like to emphasize that the authors have not used references from the MDPI journals.
There are journals like Applied Sciences, Brain Sciences, Diagnostics, International Journal of Environmental Research and Public Health, Medicina and Healthcare that have published articles in the subject area you are looking at with solid and credible approaches.
We followed PRISMA methodology for conducting this scoping review. After retrieving many studies from different databases, unfortunately we did not come across any MDPI papers.
Reviewer 3 Report
The article entitled “The Role of Neural network for detection Parkinson's Disease: A Scoping Review” is well-written and, from my point of view, would be of interest for the readers of healthcare. In spite of these, and before its publication, I would recommend that the following changes be performed:
- 2.1.1. Search sources. It is said: “s Google Scholar retrieved many articles”. I recommend detailing the exact number of articles.
- Bullet point of table 1. Please, arrange them as there is a problem in the alignament.
- In 2.4. Data extraction and data synthesis, please, explain the meaning of KD as this is not explained before.
- Table 2: please arrange table in order top put in just one page. A possible solution is splitting the table in two tables. It means that some minor changes would be requiered in the text.
- Figure 3. I would recommend removing the colour of the background in order to make the graph to seem more professional.
- Table 3: as in Table 2, please, arrange table in order top put in just one page. A possible solution is splitting the table in two tables. Please remember changing the text in order to make everything coherent.
- In 3.4.1. Public Dataset please explain the meanings of NTUA and PC-GITA.
Author Response
I would recommend that the following changes be performed:
- 1.1. Search sources. It is said: “s Google Scholar retrieved many articles”. I recommend detailing the exact number of articles.
This sentence has been deleted since it might create a misunderstanding to the reader, we included this phrase to express more appropriate meaning “We scanned only the 100 articles from Google Scholar, these articles were chosen after scanning based on their relevance to fit this paper”.
- Bullet point of table 1. Please, arrange them as there is a problem in the alignment.
Table 1 has been arranged in a proper manner.
- In 2.4. Data extraction and data synthesis, please, explain the meaning of KD as this is not explained before.
We used Two different letters for each co-author, for example MA refers to Mahmood Alzubaidi, US refers to Uzair Shah, KD refers to Khalid Dolaat. We hope that it does not create a misunderstanding in this section.
- Table 2: please arrange table in order top put in just one page. A possible solution is splitting the table in two tables. It means that some minor changes would be requiered in the text.
Table 2 has been re-arranged in order top in just one page.
- Figure 3. I would recommend removing the colour of the background in order to make the graph to seem more professional.
Background colour has been removed.
- Table 3: as in Table 2, please, arrange table in order top put in just one page. A possible solution is splitting the table in two tables. Please remember changing the text in order to make everything coherent.
Table 3 has been re-arranged in order top in just one page.
- In 3.4.1. Public Dataset please explain the meanings of NTUA and PC-GITA.
We used these synonyms as examples to dataset types used in different approaches, NTUA refers to NATIONAL TECHNICAL UNIVERSITY OF ATHENS dataset; PC-GITA refers to the research group on applied telecommunications in Universidad de Antioquia. We did not include the full terms since they are out of focus and do not bring an important information to the reader.
Round 2
Reviewer 1 Report
The authors made no effort to address my comments. Therefore, I have no choice but to stick with my original rating.
Author Response
The authors should discuss more intensively that neural networks are often a supporting technology to enable other methods (such as natural language processing). The presentation of results is not very transparent and not focused.
For scoping review, we did mention briefly about neural network but following PRISMA standard we are not required to go in details. Based on our findings highlighted all technology that have been used but we did not come across natural language processing that used separately
A discussion of diagnostic guidelines is missing.
This paper many pages and 8000 words we just mention diagnostic guidelines as traditional method to diagnose Parkinson's Disease
The selection of literature portals is insufficient because many are missing, such as Wiley, SpringerLink, Taylor & Francis, Emerald, SAGE, Web of Science).
We have scanned most common database and we obtained 91 studies as final result, this number is sufficient to conduct this review, in future we may conduct other scoping review by scanning the mentioned database.
Overall I think reviewer 1 may not in the same field